# Natural field diagnosis and molecular confirmation of fungal and bacterial watermelon pathogens in Bangladesh: A case study from the Natore and Sylhet districts

Raihan Ferdous 👁 *

Department of Plant Pathology, Sher-e-Bangla Agricultural University, Dhaka, Bangladesh

* raihanf.agri-2011106@sau.edu.bd

**Data Availability Statement:** All relevant data are within the manuscript and its Supporting Information files.

## Abstract

The study investigated watermelon diseases characterized by contrasting climatic conditions in the Sylhet and Natore Districts of Bangladesh. Sylhet experiences lower temperatures and high rainfall, while Natore has higher temperatures and low rainfall. In these survey regions, 40 watermelon fields were selected, and 10 diseases were observed, including 4 fungal, 3 bacterial, 2 water mold, and 1 viral disease. The observed diseases were Anthracnose, Cercospora leaf spot, Fusarium wilt, Gummy stem blight, Downy mildew, Phytophthora fruit rot, Bacterial fruit blotch, Angular leaf spot, Yellow vine, and Watermelon mosaic disease. Molecular analysis was done in the Plant Pathology Lab at Sher-e-Bangla Agricultural University using the specific primers for fungal (ITS1/ITS4) and bacterial (27F/1492R) DNA regions and identified nine pathogen species, excluding the causal organism of the viral disease. The identified pathogens included *Colletrotrichum orbiculare*, *Cercospora citrullina*, *Fusarium oxysporum*, *Stagonosporopsis cucurbitacearum*, *Pseudoperonospora cubensis*, *Phytophthora capsici*, *Acidovorax citrulli*, *Pseudomonas syringae*, and *Serratia marcescens*. The sequencing of the identified pathogens revealed high homology (98.91–99.71%) with known sequences in the GenBank database. Phylogenetic analysis showed six clusters for fungal and water mold pathogen isolates and three for bacterial isolates where the percentages of replicate trees were 100% in all the cases. Among the identified diseases, the highest disease occurrence was caused by Fusarium wilt (47.5%) followed by Gummy stem blight (41.5%) in the Sylhet region and Angular leaf spot (37.5%) followed by Yellow vine (33%) in the Natore area. Fusarium wilt also has a high disease intensity, demonstrating its devastating impact on yield. This study highlights the influence of environmental conditions on disease incidence and underscores the need for tailored management strategies. These findings provide a foundation for developing targeted disease management practices for sustainable watermelon cultivation in Bangladesh.

**Funding:** The author(s) received no specific funding for this work.

**Competing interests:** The author has declared that no competing interests exist.

## Introduction

Bangladeshi people love to eat watermelon (*Citrullus lanatus*) in the summer season. This significant horticulture crop is cultivated in warm locations across the globe and is prized for its juicy, sweet fruit [1]. In Bangladesh, watermelon is a major summer cash crop with high domestic demand. Farmers receive the majority of their income from the marketing and distribution of watermelon [2]. Bangladesh is seeing a daily increase in demand for watermelon consumption. In earlier decades, the state of production also improved. At present, the production of this fruit was 550,000 and 556,000 metric tons in 2021–2022 and 2022–2023 respectively [3]. It is currently produced commercially in Bangladesh, where growers can trade it for a substantial amount of foreign exchange. Therefore, watermelon production can be vital to the nation's economic growth [4].

Like those of other crops, the control of diseases and pests are major factor in watermelon production. In particular, diseases have a significant impact on crop quality and yield. Several fungal, bacterial, viral, and nemic diseases can affect watermelon. These include Angular leaf spot (*Pseudomonas syringae*), Bacterial fruit blotch (*Acidovorax citrulli*), Bacterial leaf spot (*Xanthomonas campestris*), Bacterial soft rot (*Erwinia carotovora*), Alternaria leaf spot/blight (*Alternaria cucumerin*), Anthracnose (*Colletotrichum orbiculare*), Phytophthora fruit rot (*Phytophthora capsici*), Downy mildew (*Pseudoperonospora cubensis*), Fusarium wilt (*Fusarium oxysporum*), Fusarium fruit rot (*Fusarium equiseti*), Gummy stem blight (*Stagonosporopsis cucurbitacearum*), and so on [5, 6]. When crops become infected, Fusarium wilt and Phytophthora fruit rot are the most destructive diseases to watermelon. Furthermore, foliar diseases that affect the crop annually, like Powdery mildew, Gummy stem blight, and Anthracnose, force growers to invest extensively in crop protection and agricultural practices for managing these diseases [7].

At Gurudashpur and Baraigram in Natore, farmers cultivate watermelon as an intercrop with garlic [8]. It grows widely throughout the region from April to June, and with the right care, it can be grown all year round [9]. On the other hand, at Jaintiapur and Gowainghat in Sylhet district, farmers produce watermelon as an early variety to protect their crops from flooding. They get comparatively high values as an off-season product. After receiving significant benefits, farmers are becoming more interested in watermelon farming in their region [10]. These two districts of Bangladesh are now increasingly contributing to watermelon production and uncovering the promise of exporting watermelon abroad. However, several watermelon diseases reduce the yield and quality of these fruits hampering their potential export market. Therefore, the main aims of this study are to identify the diseases of watermelons with their causal organisms and to evaluate their prevalence.

## Materials and methods

### Study site

A case study on watermelon diseases was conducted across eight blocks of four Upazilas in the Sylhet and Natore Districts of Bangladesh in 2024. These two districts were considered due to their contrasting distinct characteristics in climatic conditions and soil properties [11, 12]. Sylhet had the lowest temperature and highest rainfall while Natore had the lowest rainfall and high temperature (Table 1). However, molecular analysis was performed in the Plant Pathology Lab at Sher-e-Bangla Agricultural University in Dhaka.

### Data collection

In total, 40 fields from 8 blocks (5 fields from each block) were examined and disease symptoms, affected plant parts, intensity, status, and disease occurrence percentage were recorded.

**Table 1. Field survey locations and environmental conditions for watermelon disease assessment in Bangladesh, 2024.**

| Block name | Upazila name | District name | Visiting time | Climatic conditions (monthly average) | Soil conditions |
|---|---|---|---|---|---|
| 1. Jaintiapur-2<br>2. Sarighat-1<br>3. Alirgaon-2<br>4. Alirgaon-3 | 1. Jaintiapur<br><br>2. Gowainghat | 1. Sylhet | February, 2024 | **Temperature:** 15.1–29.5˚C<br>**Rainfall:** 35 mm<br>**Relative humidity:** 50.2% | **Soil type:** Heavy silty clay loams & clay<br>**Organic matter:** Medium<br>**Fertility level:** Medium<br>**Soil pH:** Acidic to neutral |
| 5. Upolsahar<br>6. Royna<br>7. Shidhuli<br>8. Dharabarisha | 3. Baraigram<br><br>4. Gurudashpur | 2. Natore | March, 2024 | **Temperature:** 22.4–35.1˚C<br>**Rainfall:** 24 mm<br>**Relative humidity:** 35.8% | **Soil type:** Silt loams & silty clay loams<br>**Organic matter:** Medium to high<br>**Fertility level:** Medium<br>**Soil pH:** Alkaline |

After visiting each watermelon field, 10 plants were randomly selected (for a total of 400 plants from 40 fields) to evaluate the disease occurrence. Disease occurrence was calculated employing the subsequent formula [13]:

$$Disease\ occurrence\ (\%) = \frac{Number\ of\ plants\ infected}{Number\ of\ plants\ inspected} \times 100$$

## Sample collection

Watermelon samples showing clear symptoms of fungal and bacterial infections were collected from surveyed fields in the Sylhet and Natore regions. The infected plant parts, including leaves, fruits, twigs, and stems, were carefully selected to represent a wide range of symptomatic variations. The collection was done during the growing season (Kharif season) when disease symptoms were most evident.

Each sample was immediately placed in sterile zipper bags to prevent contamination, ensuring that only the disease-causing organisms present in the field were retained. The samples were then stored in an insulated ice box to maintain a controlled temperature (4–6˚C) and prevent the degradation or overgrowth of pathogens during transportation.

Each zipper bag was properly labeled with essential information such as the sample's collection date, field location (including the block area and Upazila name), and specific plant parts. The labeled samples were then transferred to the Plant Pathology Laboratory for further analysis.

## Lab experiment

In the laboratory, the samples were stored at 4˚C. The tissue planting method involved careful slicing of plant sections into small fragments, followed by rinsing and surface sterilization with 1% Mercuric Chloride (HgCl2). The sterilized fragments were subsequently placed on Potato Dextrose Agar (PDA) medium and incubated for 6–7 days at 25±2˚C. After that, the fungal and bacterial colonies were transferred to new PDA and NA (Nutrient Agar) media respectively, to establish pure cultures. The obtained pure cultures were stored at -20˚C in a refrigerator for further analysis.

## Molecular analysis

The molecular detection process at the Plant Pathology Laboratory comprised DNA extraction, quantification, PCR amplification, gel electrophoresis with documentation, DNA purification, sequencing, and bioinformatics analysis.

## Extraction of genomic DNA

For fungi, DNA was extracted from plant tissues directly, infected by fungi and obligate pathogen viz. *Pseudoperonospora cubensis* (cannot be cultured on artificial media). In this step, the sample with a large lesion, and several conidia, mycelia, or sporangia were selected to extract DNA following the cetyltrimethylammonium bromide (CTAB) method as described in [14].

For bacteria, overnight cultures on NA media were performed in the conventional and widely used method of DNA isolation based on phenol-chloroform extraction using centrifugation at $15,000 \times g$ (Model: Z-216 M, HERMLE, Germany) as described previously [15, 16]. The obtained DNA pellets were then stored at -20˚C before further analysis.

## DNA quantification

The NanoDrop Spectrophotometer (Model: ND2000, Thermo Scientific, USA) assessed DNA purity and concentration by measuring absorbance at 260/280 nm [17].

## Polymerase chain reaction

For fungi, PCR amplification of the ITS1 region utilized primers ITS1 (5′-TCCGTAGGTGA ACCTGCGG-3′) and ITS4 (TCCTCCGCTTATTGATATGC-3′) [18]. Reactions were conducted in a Bio-Rad thermocycler (S 1000TM), with an initial denaturation step of 5 minutes at 94˚C, followed by 40 cycles of denaturation at 94˚C for 1 minute, primer annealing at 53˚C for 45 seconds, and primer extension at 72˚C for 90 seconds. The reaction mix contained 25 μL, including 2 μL (20 ng/ml) DNA templates, 12.5 μL master mix, and 10 pmol of each primer.

In the case of bacteria, the 16S ribosomal RNA (rRNA) gene was amplified using the bacteria-specific primer 27F (5′-AGAGTTGATCCTGGCTCAG-3′) and universal primer 1492R (5′-GGTTACCTTGTTACGACTT-3′), generating 1465-bp amplicons [19]. The PCR reaction mixture, totaling 25 μL, included 12.5 μL of GoTaq™ G2 hot start master mix, 1 μL of DNA (25–65 ng/μl), 1 μL of Primer 27F (10–20 pMol), 1 μL of Primer 1492R (10–20 pMol), and 9.5 μL of nuclease-free water [20]. PCR was conducted in a Gene Atlas automated thermal cycler (Model: G2) with an initial denaturation at 95˚C for 5 min, followed by 30 cycles at 95˚C for 30 s, 52˚C for 45 s, and 72˚C for 90 s, with a final extension at 72˚C for 10 min, and an overnight hold at 4˚C [21]. A 5 μL aliquot of each PCR product was electrophoresed on a 1% agarose gel and stored at -20˚C.

## Electrophoresis and gel documentation

Agarose powder (Cat: V3125) was dissolved in Tris Borate EDTA (TBE) buffer (Cat: V4251) at 80˚C for 5 min. Ethidium bromide (Cat: H5041) was added for DNA binding [21]. The gel was cast onto a Horizontal Gel Electrophoresis apparatus (Model: Mini, CBS Scientific, USA). DNA samples and a 1 kb ladder (Cat: G754B) were loaded into wells. After electrophoresis at 90 volts for 30 min, the DNA fragments were separated by size. Gel documentation was performed using an Alpha Imager Gel Documentation System (Model: Mini, Protein Simple, USA) for visualization and image capture.

## DNA purification

DNA bands from the gel were dissolved in Membrane Binding Solution, mixed with PCR amplification products, and subsequently transferred to SV Minicolumns for binding. After the samples were subsequently washed with a Membrane Wash Solution containing ethanol, the impurities were removed. After ethanol evaporation, elution was performed by

transferring the Minicolumns to clean tubes, adding nuclease-free water, and centrifuging to obtain purified DNA for further applications.

### DNA sequencing

Purified DNA samples were subsequently sent to Bioneer (Seoul, Korea) for partial sequencing via Sanger sequencing analysis. The resulting Sanger sequences were processed using Chromas 2.6. software to generate a FASTA file containing the partial sequence.

### Sequence analysis

The FASTA files of all the obtained sequences were analyzed through nucleotide BLAST by submitting the sequences in the NCBI database (https://blast.ncbi.nlm.nih.gov/Blast.cgi) to match with existing sequences in the GenBank and obtained accession numbers. After that, the evolutionary history was inferred by using the Maximum Likelihood method and the Tamura-Nei model [22]. The bootstrap consensus tree inferred from 1000 replicates was constructed to represent the evolutionary history of the taxa analyzed [23]. Branches corresponding to partitions reproduced in less than 50% of bootstrap replicates were collapsed. Initial tree (s) for the heuristic search were obtained automatically by applying Neighbor-Join and BioNJ algorithms to a matrix of pairwise distances estimated using the Tamura-Nei model and then selecting the topology with superior log likelihood value. Codon positions included were 1st +2nd+3rd+Noncoding. These analyses involved 18 nucleotide sequences of fungal and water mold and 9 sequences of bacteria. Evolutionary analyses were performed in MEGA11 software [24].

### Ethical approval

A total of 40 fields were permitted for collection of plant-infected parts samples. Department of Agricultural Extension, Govt. of Bangladesh granted the permission to perform this project. Sample collection was done with the permission and presence of all the corresponding farmers in their fields.

## Results

### Identification of diseases

During the survey, ten types of watermelon diseases were observed visually and initially confirmed by their distinctive characteristic symptoms described in (Table 2). Among the identified diseases were four fungal, three bacterial, two water mold, and one viral disease. All of the symptoms were observed above the ground parts of the watermelon plant especially leaves, fruits, stems, etc. shown in (Fig 1).

### Identification of the causal organisms

A molecular approach was performed to identify the causal organisms of watermelon diseases in the laboratory where nine species of the pathogens were identified except the viral pathogen. The identified pathogens were *Colletrotrichum orbiculare* isolate NatBD-6 (Anthracnose), *Cercospora citrullina* isolate NatBD-7 (Cercospora leaf spot), *Fusarium oxysporum* isolate isolate NatBD-8 (Fusarium wilt), *Stagonosporopsis cucurbitacearum* isolate SylBD-1 (Gummy stem blight), *Pseudoperonospora cubensis* isolate SylBD-2 (Downy mildew), *Phytophthora capsici* isolate SylBD-3 (Phytophthora fruit rot), *Acidovorax citrulli* isolate SylBD-4 (Bacterial fruit blotch), *Pseudomonas syringae* isolate SylBD-5 (Angular leaf spot), and *Serratia marcescens* isolate NatBD-9 (Yellow vine).

**Table 2. Symptoms of watermelon diseases were observed under field conditions in the Sylhet and Natore districts of Bangladesh.**

| Sl. no. | Name of disease | Symptoms observed | Disease type |
|---|---|---|---|
| 1. | Anthracnose | Initially, small, circular to angular, sunken, brown-black spots appeared in leaves and fruit skin. After that, spots were enlarged, coalesced, and finally cracked in the severe stage. | Fungal |
| 2. | Cercospora leaf spot | The disease is characterized by a small, circular, grey in center, dark brown margin color. The leaves were dead when the spots were enlarged while the center turned tan to grey. | Fungal |
| 3. | Fusarium wilt | Initially, the foliage appeared dull, greyish green while the vines were dry, brown colored, and died in severe cases. It showed partial wilt. A clear brown discoloration was observed in the center of the vascular core near the soil line. | Fungal |
| 4. | Gummy stem blight | The leaf spots were large, round to uneven in shape, dark brown to black colored. The spots usually were developed from the margin of the leaves. | Fungal |
| 5. | Downy mildew | The older leaves were affected first and showed a pale green to dark brown color, circular to irregular spots. As the disease advanced, leaves were curled, upright, burnt, and died. | Water mold |
| 6. | Phytophthora fruit rot | Firstly, the infection began on the side closest to the ground. The symptoms appeared as a brown to dark color, water-soaked lesion that expanded and became covered with white mold. Fruit collapsed completely as a result of the fruit rot spread.Top of Form | Water mold |
| 7. | Bacterial fruit blotch | Initially, small water-soaked lesions appeared on the upper surface of the fruit skin. Then the water-soaked lesions expanded quickly at the mature stage and cracks were observed in severe cases.Top of Form | Bacteria |
| 8. | Angular leaf spot | The lesion started as tiny brown to black spots which were angular-shaped, and in some cases not angular-shaped with or without a yellow halo. | Bacteria |
| 9. | Yellow vine | The vine's terminal leaves were curling up and the leaves had typically turned yellow. | Bacteria |
| 10. | Watermelon mosaic | The symptoms were most noticeable in young, quickly- growing plants. The leaves had light and dark green mosaic patterns and were deformed, pale green, and puckered. Fruits were dwarfed and spotted. | Virus |

## Study on PCR products

The Gel-Doc system visualized PCR products of fungal and water mold isolates generated by ITS1 and ITS4 primers, which resulted in the DNA bands of approximately 710 base pairs (bp). Variability in amplicon sizes ranging from 512 to 751 bp was observed in the GenBank database. In the case of bacterial isolates, the DNA bands of approximately 1465 bp were detected using primers 27F and 1492R where the amplicon sizes were 1367 to 1410 bp in the database. The generated DNA bands indicated the successful PCR amplification, which were extremely similar in size to the target band. The size of the amplified DNA fragment was verified using a Bench Top 1 kb DNA ladder, which acts as a size marker. Fig 2 (Lanes 1–6) shows the successful amplification of DNA bands obtained from several fungal and water mold isolates whereas Fig 3 (Lanes 1–3), displays the bacterial isolates. These confirm the specificity and successful amplification of the desired DNA fragments under the utilized primer and PCR conditions, as demonstrated by the Gel-Doc system.

## Analysis of DNA sequences

The obtained FASTA sequences were analyzed using the BLAST tool on the NCBI website (S1 Appendix). These sequences were found to match existing nucleotide sequences in the NCBI GenBank database. All the sequences exceeded 98% (98.91–99.71%) homology with their corresponding sequence. Subsequently, unique accession numbers were assigned to the partial sequences for four fungal, two water mold, and three bacterial isolates (Table 3).

## Analysis of phylogenetic tree

The phylogenetic trees were analyzed using the fungal, water mold, and bacterial sequences, where the highest log likelihood was -4119.79 for fungal and water mold sequences. Moreover, for bacterial sequences, it was log -3797.87. There were a total of 1065 and 1446

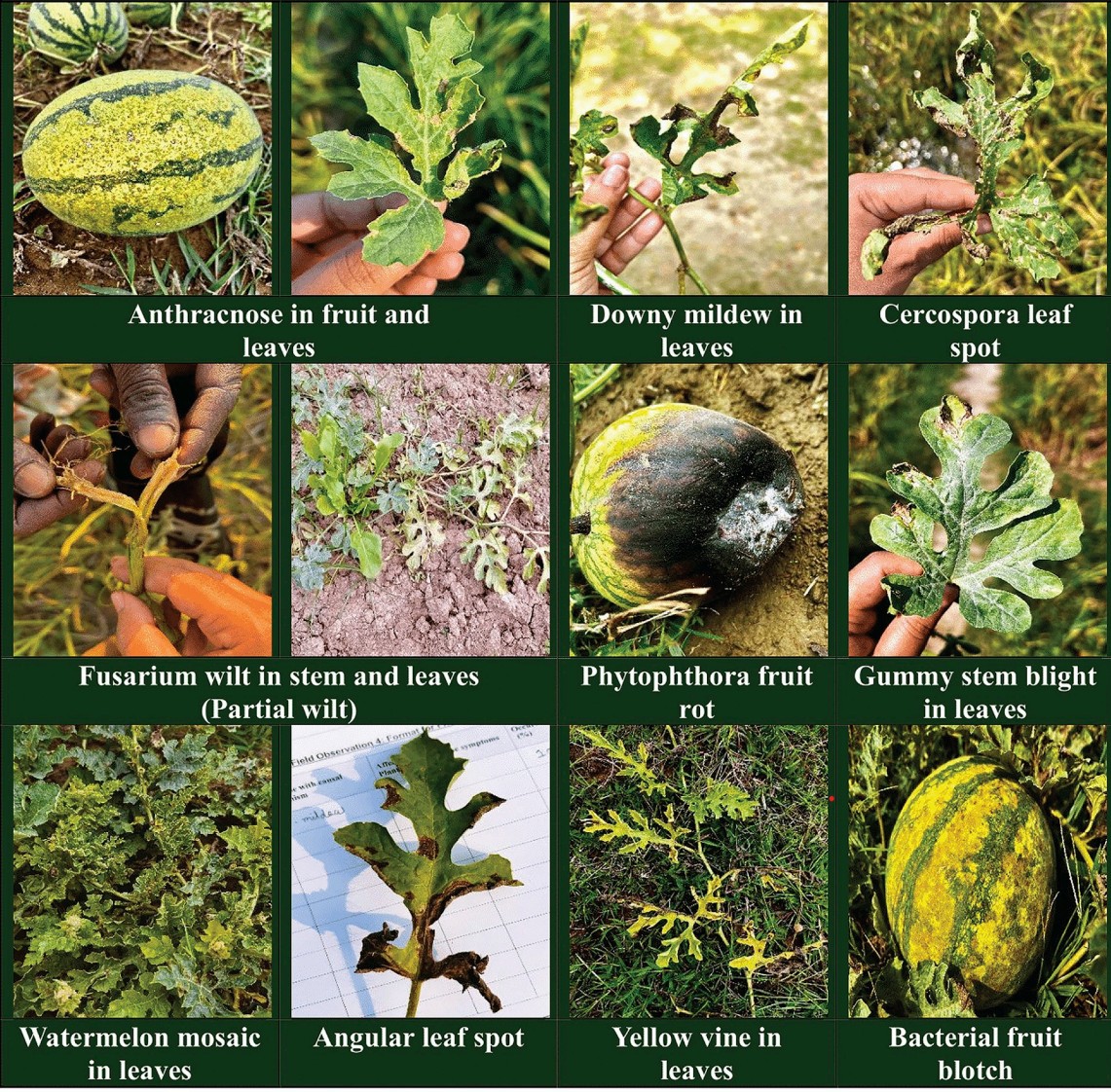

**Fig 1. Watermelon disease symptoms were observed in different infected plant parts during the field study.**

positions in the final dataset obtained from fungal and bacterial sequences respectively. Based on the phylogenetic tree, the studied strains fell into six main clusters of fungal and water mold and three main clusters of bacterial isolates. In (Fig 4), shows the clusters of *Colletrotrichum*, *Fusarium*, *Pseudoperonospora*, *Cercospora*, *Stagonosporopsis*, *and Phytophthora* where all the clusters exhibited 100% replicate trees in which the associated taxa clustered together in the bootstrap test 1000 replicates are shown next to the branches. On the other hand, (Fig 5) shows the clusters of bacterial isolates which were *Acidovorax*, *Pseudomonas*, and *Serratia* and these displayed 100% replicate trees for all the clusters. The genetic distances between the strains are also shown in the trees. The distances are measured by the number of substitutions per site. Considering the percentage of replicate trees and genetic distances, the phylogenetic tree shows that the studied strains were all relatively closely related to each other within a cluster.

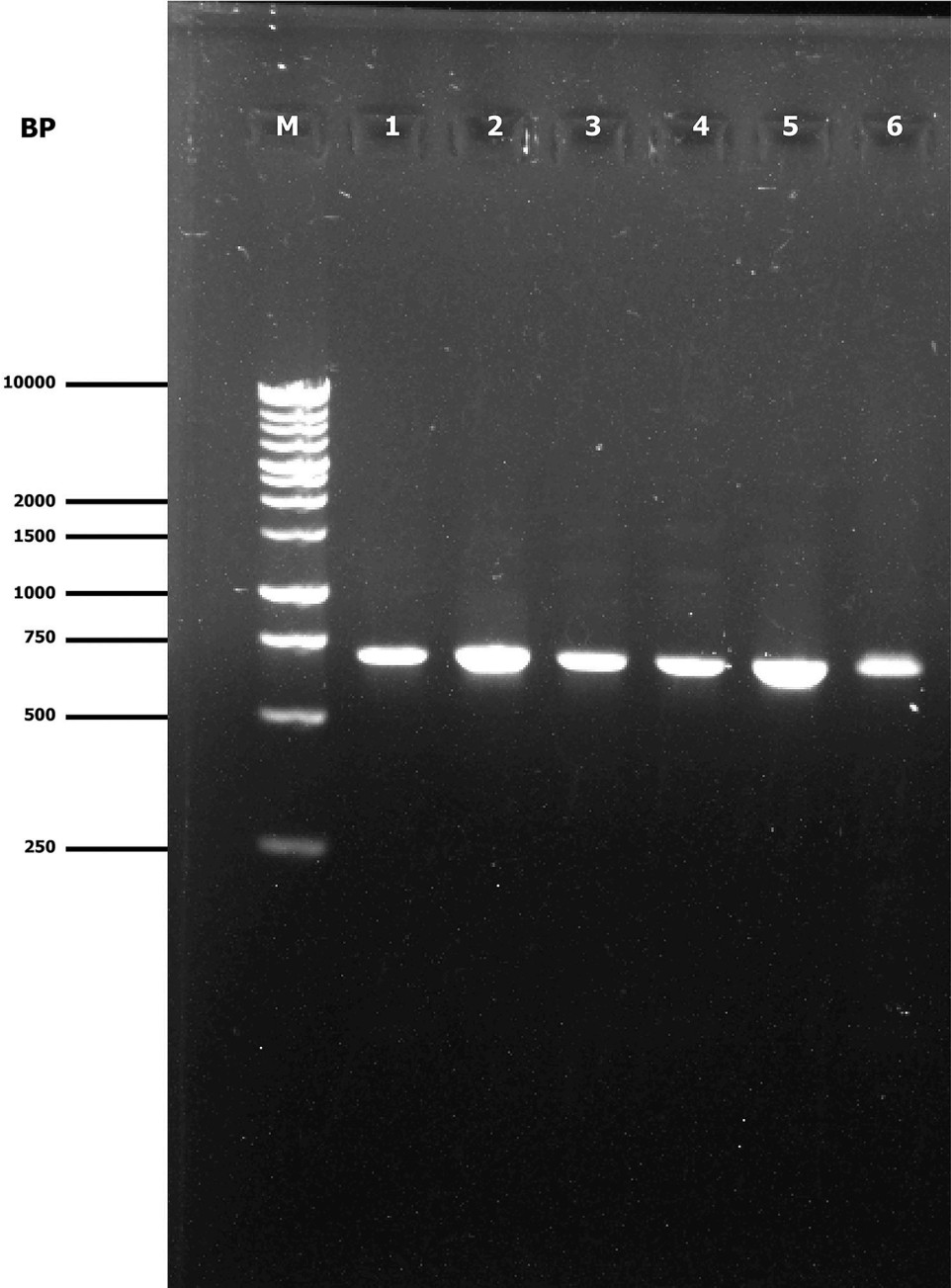

**Fig 2.** PCR amplified products of fungus and water mold obtained from (1) *Colletrotrichum orbiculare* isolate NatBD-6, (2) *Cercospora citrullina* isolate NatBD-7, (3) *Fusarium oxysporum* isolate NatBD-8, (4) *Stagonosporopsis cucurbitacearum* isolate SylBD-1, (5) *Pseudoperonospora cubensis* isolate SylBD-2, and (6) *Phytophthora capsici* isolate SylBD-3. M: denotes 1kb DNA ladder (Marker).

## Evaluation of disease status, intensity, and occurrence

During the survey, 400 plants were observed in 40 fields of 8 different blocks in the Sylhet and Natore districts to evaluate the prevalence of corresponding diseases (S2 Appendix). Among the identified diseases, the highest disease occurrence in Sylhet was Fusarium wilt (47.58%) followed by Gummy stem blight (41.5), and in Natore, it was Angular leaf spot (37.5) followed by

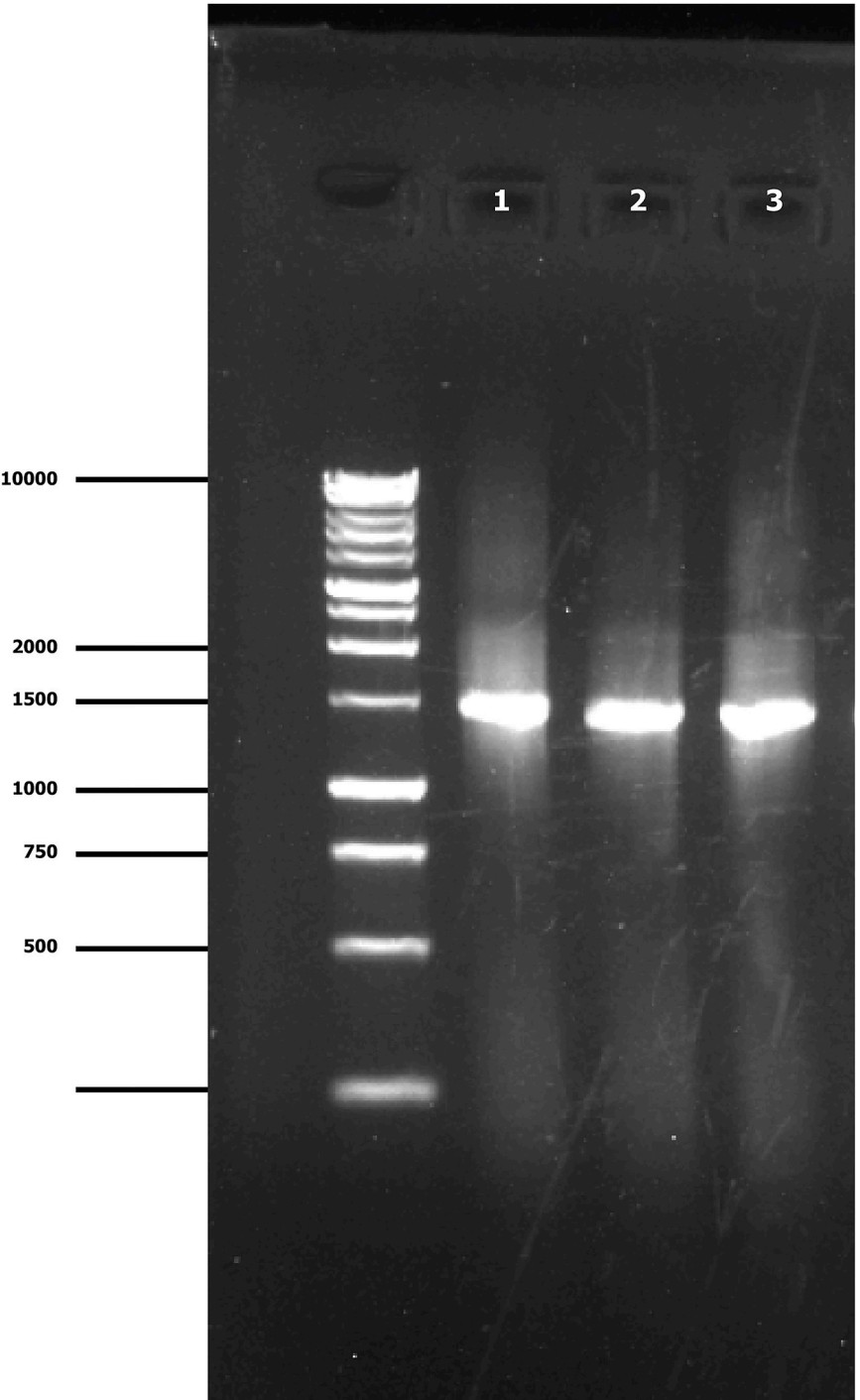

**Fig 3.** PCR amplified products of bacteria obtained from (1) *Acidovorax citrulli* isolate SylBD-4, (2) *Pseudomonas syringae* isolate SylBD-5, and (3) *Serratia marcescens* isolate NatBD-9. M: denotes 1kb DNA ladder (Marker).

Yellow vine (33%). However, Downy mildew and Watermelon mosaic diseases were completely absent in the Natore region while Cercospora leaf spot disease was not observed in the Sylhet zone. In the early growing stage of watermelon plants, Fusarium wilt showed the most intent to damage the plants and spread as a major disease while Gummy stem attacked in

**Table 3. Results of gene identification with the BLAST program.**

| LAB strain | | | BLAST Alignment | | | | |
|---|---|---|---|---|---|---|---|
| Sl. no. | Identified pathogen isolates | Obtained accession no. | Species | Query coverage | E value | Percent identity | Accession no. |
| 1. | *Colletrotrichum orbiculare* isolate NatBD-6 | PP837831 | *Colletrotrichum orbiculare* isolate POL1 | 100% | 0.0 | 99.26% | ON398802 |
| 2. | *Cercospora citrullina* isolate NatBD-7 | PP837832 | *Cercospora citrulline* isolate Cer 76–18 | 100% | 0.0 | 98.91% | OL589543 |
| 3. | *Fusarium oxysporum* isolate NatBD-8 | PP837833 | *Fusarium oxysporum* f. sp. *niveum* | 100% | 0.0 | 99.39% | FJ156282 |
| 4. | *Stagonosporopsis cucurbitacearum* isolate SylBD-1 | PP837834 | *Stagonosporopsis cucurbitacearum* isolate RKS8 | 100% | 0.0 | 99.22% | OL774657 |
| 5. | *Pseudoperonospora cubensis* isolate SylBD-2 | PP837835 | *Pseudoperonospora cubensis* isolate E1CHK0 | 100% | 0.0 | 99.18% | MW845754 |
| 6. | *Phytophthora capsici* isolate SylBD-3 | PP837836 | *Phytophthora capsici* isolate 219 | 100% | 0.0 | 99.60% | DQ464028 |
| 7. | *Acidovorax citrulli* isolate SylBD-4 | PP837828 | *Acidovorax citrulli* strain RKFB 793 | 100% | 0.0 | 99.71% | KP410333 |
| 8. | *Pseudomonas syringae* isolate SylBD-5 | PP837829 | *Pseudomonas syringae* strain BP550 | 100% | 0.0 | 99.57% | KX533931 |
| 9. | *Serratia marcescens* isolate NatBD-9 | PP837830 | *Serratia marcescens* strain QQ3 | 100% | 0.0 | 99.57% | OP861103 |

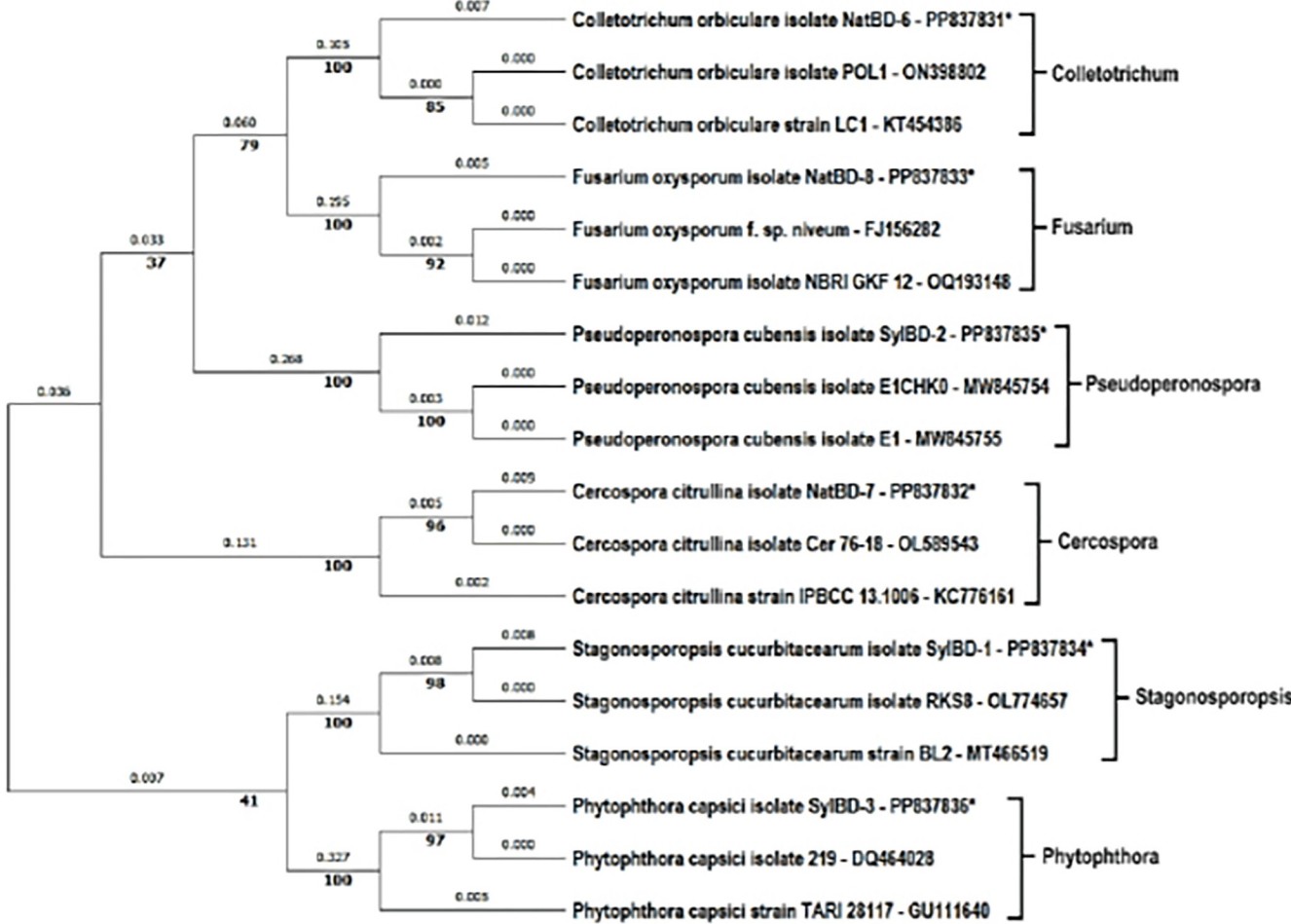

**Fig 4. Phylogenetic tree for fungal and water mold pathogens of watermelon plants collected from Natore and Sylhet regions in Bangladesh.** This analysis involved 18 nucleotide sequences where the clusters fall into 6 distinct clades of fungal and water mold strains. * Denotes the isolates obtained from the watermelon samples.

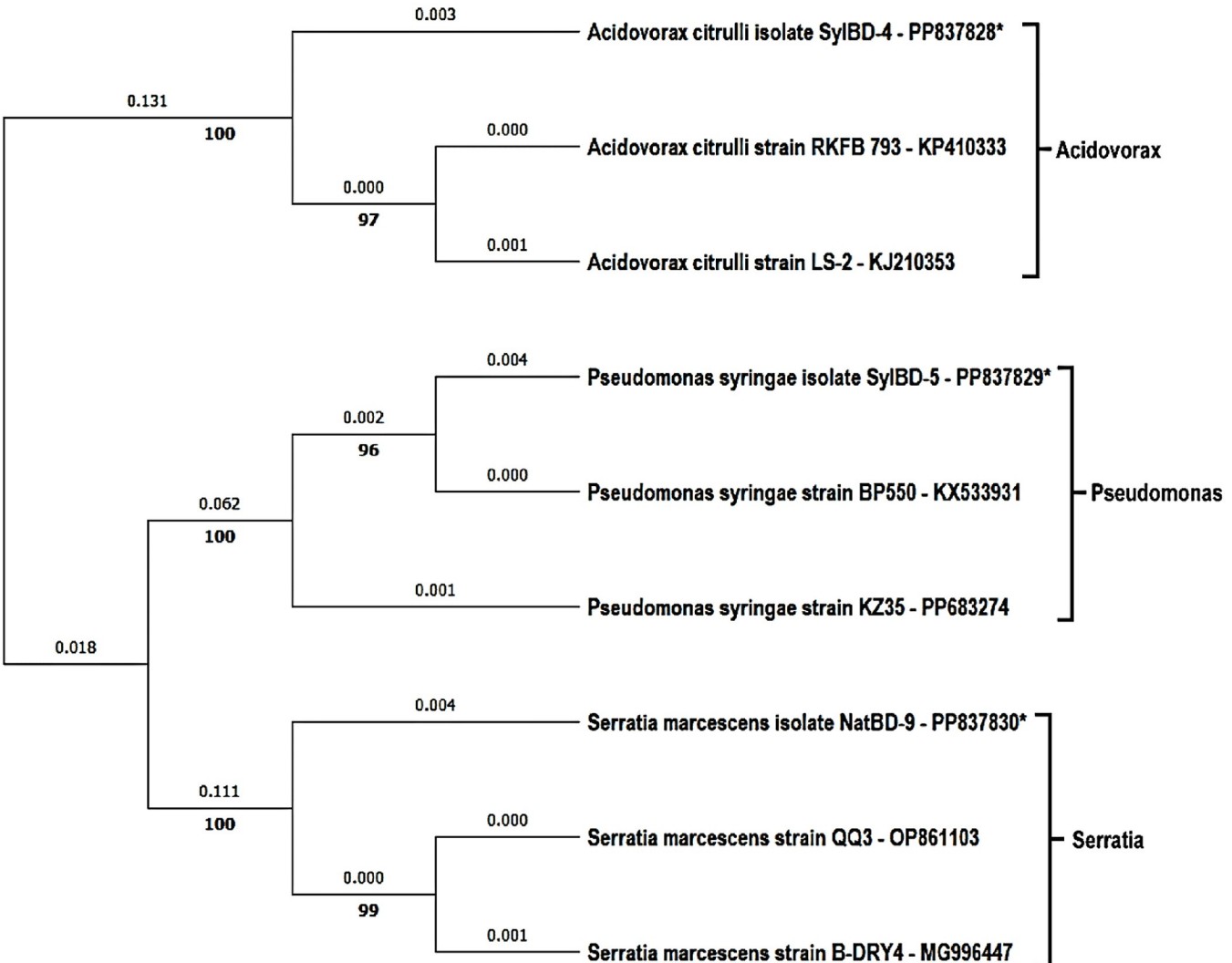

**Fig 5. Phylogenetic tree for bacterial pathogens of watermelon plants collected from Natore and Sylhet regions in Bangladesh.** This analysis involved 9 nucleotide sequences where the clusters fall into 3 distinct clades of bacterial strains. * Denotes the isolates obtained from the watermelon samples.

the late stage with medium disease intensity. However, Phytophthora fruit rot was evaluated as a minor disease but its intensity to damage the watermelon fruits was high (Table 4).

## Discussion

This study aimed to investigate watermelon diseases across two distinct agro-ecological zones in Bangladesh, Sylhet, and Natore, providing valuable insights into the prevalence of these diseases, causative agents, and molecular identification. The contrasting climatic conditions of these districts, Sylhet with its lower temperatures and high rainfall, and Natore with higher temperatures and low rainfall, served as compelling backlines for understanding disease dynamics in various environmental contexts.

### Disease prevalence

The survey identified ten diseases affecting watermelon plants, which were categorized into fungal, bacterial, water mold, and viral diseases. Notably, Fusarium wilt emerged as the most

**Table 4. Prevalence of the identified watermelon diseases in Sylhet and Natore regions.**

| Sl. no. | Observed diseases | Identified pathogens | Disease | | | | |
|---------|-------------------|---------------------|---------|---|---|---|---|
| | | | Occurrence (%) | | Status | | intensity |
| | | | Sylhet | Natore | Sylhet | Natore | |
| 1. | Anthracnose | *C. orbiculare* isolate NatBD-6 | 26 | 8.5 | Minor | Minor | Low |
| 2. | Cercospora leaf spot | *C. citrullina* isolate NatBD-7 | 0 | 10.5 | Minor | Minor | Low |
| 3. | Fusarium wilt | *F. oxysporum* isolate NatBD-8 | 47.5 | 32 | Major | Minor | High |
| 4. | Gummy stem blight | *S. cucurbitacearum* isolate SylBD-1 | 41.5 | 18.5 | Major | Minor | Medium |
| 5. | Downy mildew | *P. cubensis* isolate SylBD-2 | 32.5 | 0 | Minor | Minor | Low |
| 6. | Phytophthora fruit rot | *P. capsici* isolate SylBD-3 | 29 | 4 | Minor | Minor | High |
| 7. | Bacterial fruit blotch | *A. citrulli* isolate SylBD-4 | 3.5 | 13.5 | Minor | Minor | Low |
| 8. | Angular leaf spot | *P. syringae* isolate SylBD-5 | 17.5 | 37.5 | Minor | Minor | Low |
| 9. | Yellow vine | *S. marcescens* isolate NatBD-9 | 23.5 | 33 | Minor | Minor | Low |
| 10. | Watermelon mosaic | *Watermelon mosaic virus* | 26.5 | 0 | Minor | Minor | Medium |

prevalent disease, with an occurrence rate of 47.5%. This high prevalence aligns with global reports highlighting Fusarium wilt as a significant threat to watermelon production due to its soil-borne nature and persistent survival in soil [25]. Gummy stem blight and Angular leaf spot were also prevalent, with occurrences of 41.5% and 37.5%, respectively [26, 27]. The 2016 Pest Risk Analysis of cucurbits in Bangladesh reported that most identified diseases posed a minor impact on watermelon production. However, this study highlights a growing concern, as Fusarium wilt has now become a major threat due to its high occurrence and intensity. Additionally, Gummy stem blight, with its considerable occurrence and medium intensity, is emerging as another significant challenge. These findings indicate a shift in disease dynamics, posing a substantial risk to watermelon production in the region [28]. The survey results underscore the need for targeted management strategies, particularly for Fusarium wilt, and Gummy stem blight which exhibited the highest intensity during the growth stages of watermelon plants.

## Identification of pathogens

The molecular identification, including DNA extraction, PCR amplification, gel electrophoresis, and sequencing is crucial for accurately identifying the pathogens. The use of specific primers for fungal (ITS1/ITS4) and bacterial (27F/1492R) DNA regions ensured precise amplification of target sequences. The molecular analysis confirmed that nine distinct pathogen species were responsible for the observed diseases and the sequencing revealed high homology (98.91–99.71%) with known sequences in the GenBank database, confirming the accuracy of the pathogen identification. The pathogens identified included *Colletrotrichum orbiculare* NatBD-6 (Anthracnose), *Cercospora citrullina* NatBD-7 (Cercospora leaf spot), *Fusarium oxysporum* NatBD-8 (Fusarium wilt), *Stagonosporopsis cucurbitacearum* SylBD-1 (Gummy stem blight), *Pseudoperonospora cubensis* SylBD-2 (Downy mildew), *Phytophthora capsici* SylBD-3 (Phytophthora fruit rot), *Acidovorax citrulli* SylBD-4 (Bacterial fruit blotch), *Pseudomonas syringae* SylBD-5 (Angular leaf spot), and *Serratia marcescens* NatBD-9 (Yellow vine). The identification of these pathogens also identified by several researchers in the watermelon field provides a crucial foundation for developing specific control measures [5, 6].

## Phylogenetic analysis

The phylogenetic analysis revealed six main clusters for fungal and water mold pathogens and three for bacterial isolates, indicating the genetic relatedness among the strains. The high

bootstrap values (100%) in the phylogenetic trees confirm the reliability of these groupings, providing a robust framework for understanding the evolutionary relationships of the pathogens involved.

### Environmental influence on disease distribution

The contrasting environmental conditions of Sylhet and Natore districts significantly influenced disease distribution and prevalence. The high rainfall and cooler temperatures in Sylhet likely favored the proliferation of water molds and certain fungal pathogens, such as *Pseudoperonospora cubensis* and *Phytophthora capsica*, *Colletotrichum orbiculare*. Conversely, the hotter and drier conditions in Natore were more conducive to bacteria like *Acidovorax citrulli* and *Pseudomonas syringae*. This environmental influence underscores the necessity of region-specific disease management practices that consider local climatic conditions.

### Implications for disease management

The findings of this study have important implications for watermelon disease management in Bangladesh. The high prevalence and intensity of Fusarium wilt, particularly in the early growth stages, highlight the need for early intervention strategies, such as soil fumigation, crop rotation, and resistant cultivars. For other prevalent diseases like Gummy stem blight and Angular leaf spot, integrated pest management (IPM) approaches, including the use of fungicides, proper irrigation practices, and resistant varieties, should be emphasized. The molecular identification of pathogens also opens avenues for the development of molecular-based diagnostic tools, enabling rapid and accurate detection of these pathogens in the field.

### Conclusion

This comprehensive study provides a detailed account of the prevalence, identification, and molecular characterization of watermelon diseases in two agroecologically distinct districts of Bangladesh. The findings highlight the significant impact of environmental conditions on disease distribution and underscore the need for tailored disease management strategies. Future research should focus on developing and implementing integrated disease management practices that consider local environmental conditions and pathogen characteristics. Such efforts will be crucial in mitigating the impact of these diseases on watermelon production and ensuring sustainable agricultural practices in the region.

### Supporting information

**S1 Appendix. FASTA nucleotide sequences of the identified fungal and bacterial isolates.**
(PDF)

**S2 Appendix. Evaluation of disease occurrence in the infected fields.** ¾
(XLSX)

### Acknowledgments

The author would like to thank the Center for Resource Development Studies Ltd. (CRDS) for allowing him to do the field survey in the Sylhet and Natore regions.

### Author Contributions

**Conceptualization:** Raihan Ferdous.

**Data curation:** Raihan Ferdous.

**Formal analysis:** Raihan Ferdous.

**Funding acquisition:** Raihan Ferdous.

**Investigation:** Raihan Ferdous.

**Methodology:** Raihan Ferdous.

**Project administration:** Raihan Ferdous.

**Resources:** Raihan Ferdous.

**Software:** Raihan Ferdous.

**Validation:** Raihan Ferdous.

**Visualization:** Raihan Ferdous.

**Writing – original draft:** Raihan Ferdous.

**Writing – review & editing:** Raihan Ferdous.

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
