## [Decision Letter · Decision Letter 0]

20 Sep 2024

PONE-D-24-26919Natural field diagnosis and molecular confirmation of fungal and bacterial watermelon pathogens in Bangladesh: A case study from Natore and Sylhet districtPLOS ONE

Dear Dr. Ferdous,

Thank you for submitting your manuscript to PLOS ONE. After careful consideration, we feel that it has merit but does not fully meet PLOS ONE’s publication criteria as it currently stands. Therefore, we invite you to submit a revised version of the manuscript that addresses the points raised during the review process.

We look forward to receiving your revised manuscript.

Kind regards,

Kandasamy Ulaganathan

Academic Editor

PLOS ONE

Journal Requirements:

2. We suggest you thoroughly copyedit your manuscript for language usage, spelling, and grammar. If you do not know anyone who can help you do this, you may wish to consider employing a professional scientific editing service. The American Journal Experts (AJE) (https://www.aje.com/) is one such service that has extensive experience helping authors meet PLOS guidelines and can provide language editing, translation, manuscript formatting, and figure formatting to ensure your manuscript meets our submission guidelines. Please note that having the manuscript copyedited by AJE or any other editing services does not guarantee selection for peer review or acceptance for publication. Upon resubmission, please provide the following: ● The name of the colleague or the details of the professional service that edited your manuscript ● A copy of your manuscript showing your changes by either highlighting them or using track changes (uploaded as a *supporting information* file) ● A clean copy of the edited manuscript (uploaded as the new *manuscript* file)

5. PLOS ONE now requires that authors provide the original uncropped and unadjusted images underlying all blot or gel results reported in a submission’s figures or Supporting Information files. This policy and the journal’s other requirements for blot/gel reporting and figure preparation are described in detail at https://journals.plos.org/plosone/s/figures#loc-blot-and-gel-reporting-requirements and https://journals.plos.org/plosone/s/figures#loc-preparing-figures-from-image-files. When you submit your revised manuscript, please ensure that your figures adhere fully to these guidelines and provide the original underlying images for all blot or gel data reported in your submission. See the following link for instructions on providing the original image data: https://journals.plos.org/plosone/s/figures#loc-original-images-for-blots-and-gels. In your cover letter, please note whether your blot/gel image data are in Supporting Information or posted at a public data repository, provide the repository URL if relevant, and provide specific details as to which raw blot/gel images, if any, are not available. Email us at plosone@plos.org if you have any questions.

Reviewers' comments:

Reviewer's Responses to Questions

**Comments to the Author**

1. Is the manuscript technically sound, and do the data support the conclusions?

Reviewer #1: Yes

Reviewer #2: Yes

2. Has the statistical analysis been performed appropriately and rigorously? 

Reviewer #1: I Don't Know

Reviewer #2: N/A

3. Have the authors made all data underlying the findings in their manuscript fully available?

Reviewer #1: Yes

Reviewer #2: Yes

4. Is the manuscript presented in an intelligible fashion and written in standard English?

Reviewer #1: No

Reviewer #2: Yes

5. Review Comments to the Author

Reviewer #1: The study highlights the prevalence of different fungal and bacterial pathogens of watermelon and their identification through molecular techniques. The study investigated watermelon diseases in Sylhet and Natore Districts of Bangladesh. In these survey regions, out of 40 watermelon fields, 10 diseases were observed, including 4 fungal, 3 bacterial, 2 water mold, and 1 viral disease. The observed diseases were Anthracnose, Cercospora leaf spot, Fusarium wilt, Gummy stem blight, Downy mildew, Phytophthora fruit rot, Bacterial fruit blotch, Angular leaf spot, Yellow vine, and Watermelon mosaic disease.

However, the study lacks novelty and is of local importance.

Reviewer #2: The manuscript need revision.

1. Keywords section, remove one keyword

2. In the introduction section, add one paragraph with the latest citation.

3. Add the reference PMID: 35867657

4. Add few Reference in the reference section of the manuscript.

5. Discuss more in the Discussion section of the manuscript with citations.

6. If the author provide the samples location then it will be better for the study.

7. Describe the details of the sample collection methods.

8. Why the author use Maximum Likelihood method? why not NJ method?

9. Check the grammatical mistake throughout the manuscript.

6. PLOS authors have the option to publish the peer review history of their article (what does this mean?). If published, this will include your full peer review and any attached files.

Reviewer #1: **Yes: **Dr KS Hooda

Reviewer #2: **Yes: **Dr. Ajaya Kumar Rout

---

## [Author Response · Author response to Decision Letter 0]

25 Sep 2024

Date: 26-09-2024

Academic Editor,

PLOS ONE

Dear Kandasamy Ulaganathan,

The author would like to thank you and the reviewers for your thoughtful and constructive feedback on our manuscript [PONE-D-24-26919]. The author greatly appreciates the time and effort put into reviewing our work, and we have carefully considered each point raised. In response, the author has made the necessary revisions and clarifications throughout the manuscript to address all concerns.

Enclosed is a table detailing each comment made by the academic editor and reviewers, alongside author responses and the corresponding revisions. The author believes these changes have strengthened the manuscript, and we hope that it now meets the journal’s standards.

Academic editor:

Response by Author: We have ensured that the manuscript is now fully formatted according to PLOS ONE's style requirements, including file naming conventions. If there are any specific areas that require further adjustment, please let us know, and we will address them accordingly.

2. We suggest you thoroughly copyedit your manuscript for language usage, spelling, and grammar. 

Response by Author: We have thoroughly reviewed the manuscript, enhancing the language, spelling, and grammar through careful observation. The text has been edited for clarity, consistency, and readability. If there are any specific sections where further improvements are needed, kindly point them out, and we will address them immediately.

3. Your ethics statement should only appear in the Methods section of your manuscript. 

Response by Author: The ethics statement has been moved to the Methods section as per the journal's guidelines.

4. Please include captions for your Supporting Information files at the end of your manuscript, and update any in-text citations to match accordingly. 

Response by Author: Captions for the Supporting Information files have been added at the end of the manuscript, and all in-text citations have been updated to match accordingly.

5. PLOS ONE now requires that authors provide the original uncropped and unadjusted images underlying all blot or gel results reported in a submission’s figures or Supporting Information files.

Response by Author: We have ensured that all figures adhere to PLOS ONE's guidelines. The original, uncropped, and unadjusted images underlying all blot or gel data reported in the manuscript have been provided as required.

Reviewer #1 

1. The study lacks novelty and is of local importance. 

Response by Author: We appreciate the reviewer's feedback regarding the novelty and local importance of the study. While we acknowledge that the study focuses on a specific region (Bangladesh), it addresses a critical issue from a quarantine and agricultural export perspective. Identifying and documenting potential diseases in watermelon is crucial for the Bangladeshi government and regulatory authorities to ensure food safety, prevent disease outbreaks, and protect local agriculture.

Furthermore, this research holds international relevance, as it provides critical insights to global stakeholders, particularly those involved in importing watermelon from Bangladesh. By identifying any new or emerging watermelon diseases in the region, we offer valuable information that can help importing countries implement appropriate quarantine measures and protect their agricultural sectors from potential threats. Therefore, while rooted in a local context, the findings have wider implications for international trade, biosecurity, and disease management.

We hope this clarification emphasizes the significance of the study beyond its local scope and provides a better understanding of its broader impact.

Response to Reviewer #2 

1. Keywords section, remove one keyword.

Response by Author: We appreciate the reviewer's suggestion. The keyword "PCR" has been removed from the Keywords section as requested.

2. In the introduction section, add one paragraph with the latest citation. 

Response by Author: We appreciate the reviewer's suggestion to add a paragraph with the latest citation. Upon reviewing the Introduction section, we believe that all current citations are both relevant and up-to-date. However, if there are specific points or topics you feel should be further elaborated upon, we would welcome your suggestions to ensure the introduction fully addresses all relevant aspects of the study.

Additionally, if you have identified any citations that may no longer reflect the most recent findings, we would be happy to replace them with more current references upon your recommendation. We value your input and are committed to ensuring the highest quality and accuracy in our work.

3. Add the reference PMID: 35867657.

Response by Author: We appreciate the reviewer's suggestion to add the reference PMID: 35867657. However, upon reviewing the paper titled "Structural insights into the RNA interaction with Yam bean Mosaic virus (coat protein) from Pachyrhizus erosus using bioinformatics approach," we found that this study focuses on RNA interaction with viral diseases, which is not directly relevant to our research focus.

4. Add few Reference in the reference section of the manuscript. 

Response by Author: Thank you for your suggestion to add additional references in the reference section. Currently, there are 27 references included in the reference section, all of which have been cited within the manuscript.

If you have identified any citations that are missing from the reference list or if there are specific references that you believe should be added to strengthen the manuscript, kindly indicate the corresponding reference number(s). We would be happy to make the necessary additions or corrections based on your feedback

5. Discuss more in the Discussion section of the manuscript with citations. 

Response by Author: We have added additional essential points to the Discussion section, along with relevant citations, to provide a more comprehensive analysis of the study's findings. These new points address the emerging threats posed by Fusarium wilt and Gummy stem blight in watermelon production and their potential impact, referencing recent studies that support our observations. We hope this expanded discussion meets the reviewer's expectations.

6. If the author provide the samples location then it will be better for the study. 

Response by Author: The sample locations, including block areas and Upazila names, have been provided in Table 1. Additionally, the table includes relevant information on these locations' climatic and soil conditions to give a more comprehensive context for the study. We hope this adequately addresses your concern and enhances the clarity of the study.

7. Describe the details of the sample collection methods.

Response by Author: The sample collection methods have now been described in detail in the revised manuscript. The updated section provides comprehensive information on the process, including how symptomatic plant parts were selected, stored, labeled, and transported, as well as the steps taken to ensure the integrity of the samples during collection and laboratory analysis. We hope this addresses your concerns and enhances the clarity of our methodology.

8. Why the author use Maximum Likelihood method? why not NJ method? 

Response by Author: We appreciate the reviewer’s inquiry regarding the use of the Maximum Likelihood (ML) method over the Neighbor-Joining (NJ) method in our analysis. The decision to use the ML method was based on its advantages in providing more robust and accurate phylogenetic estimations.

The NJ method, while faster and computationally less demanding, is a distance-based approach and does not account for the probabilistic nature of evolutionary changes, which can lead to less precise results, particularly for datasets with high variability. For our study, the higher precision and reliability offered by the ML method were essential for producing accurate phylogenetic inferences, which is why it was chosen for the MEGA analysis.

9. Check the grammatical mistake throughout the manuscript. 

Response by Author: Thank you for your observation. We have thoroughly reviewed the manuscript multiple times and addressed any grammatical issues that were identified. If there are specific sections where you notice errors or believe further improvements are needed, kindly point them out, and we will make the necessary revisions. We are committed to ensuring the manuscript is grammatically sound.

We look forward to your further evaluation of our revised manuscript.

Sincerely,

Raihan Ferdous

Corresponding author

---

## [Decision Letter · Decision Letter 1]

23 Oct 2024

Natural field diagnosis and molecular confirmation of fungal and bacterial watermelon pathogens in Bangladesh: A case study from the Natore and Sylhet districts

PONE-D-24-26919R1

Dear Dr. Ferdous,

We’re pleased to inform you that your manuscript has been judged scientifically suitable for publication and will be formally accepted for publication once it meets all outstanding technical requirements.

Kind regards,

Kandasamy Ulaganathan

Academic Editor

PLOS ONE

Additional Editor Comments (optional):

Reviewers' comments:

Reviewer's Responses to Questions

**Comments to the Author**

1. If the authors have adequately addressed your comments raised in a previous round of review and you feel that this manuscript is now acceptable for publication, you may indicate that here to bypass the “Comments to the Author” section, enter your conflict of interest statement in the “Confidential to Editor” section, and submit your "Accept" recommendation.

Reviewer #2: All comments have been addressed

2. Is the manuscript technically sound, and do the data support the conclusions?

Reviewer #2: Yes

3. Has the statistical analysis been performed appropriately and rigorously? 

Reviewer #2: N/A

4. Have the authors made all data underlying the findings in their manuscript fully available?

Reviewer #2: Yes

5. Is the manuscript presented in an intelligible fashion and written in standard English?

Reviewer #2: Yes

6. Review Comments to the Author

Reviewer #2: The manuscript is revised. The author addresses all the queries. My decision is to accept the manuscript for publication.

7. PLOS authors have the option to publish the peer review history of their article (what does this mean?). If published, this will include your full peer review and any attached files.

Reviewer #2: **Yes: **Dr. Ajaya Kumar Rout

---

## [Editor Report · Acceptance letter]

28 Oct 2024

PONE-D-24-26919R1 

PLOS ONE

Dear Dr. Ferdous, 

I'm pleased to inform you that your manuscript has been deemed suitable for publication in PLOS ONE. Congratulations! Your manuscript is now being handed over to our production team.

Kind regards, 

on behalf of

Dr. Kandasamy Ulaganathan 

Academic Editor

PLOS ONE